# DIOMIX: A Dynamic Multi-Agent Reinforcement Learning Mixing Structure for Independent Intra-Option Learning

## Abstract

In cooperative multi-agent reinforcement learning (MARL), agents are equipped with a formalism to plan, learn, and reason in diverse ways, enabling continual knowledge accumulation over time. Each agent must consistently learn within its environment and possess the ability to reason at various levels of both temporal and spatial abstraction to navigate the intricacies specific to its surroundings. Current state-of-the-art approaches explicitly rely on learning an objective function that harmonizes both planning and learning without explicitly relying on reasoning. We propose a distinctive framework, Dynamic Intra-Options Mixtures (DIOMIX), aiming to address the deficiency in reasoning capabilities present in current state-of-the-art algorithms. We introduce an agent-independent option-based framework, incorporating a notion of temporal abstraction into the MARL paradigm using an advantage-based learning scheme directly on the option policy. This scheme enables higher long-term utility retention compared to directly optimizing action-value functions themselves. However, using temporal difference learning could hinder the optimization of extended temporal actions; therefore, to mitigate this issue where options are optimized solely to execute as primitive actions, we incorporate a regularization mechanism into the learning process to enable options execution over extended periods. Through quantitative and qualitative empirical results, DIOMIX can acquire individually separable and explainable reasoning capabilities that lead to agent specialization, task simplification, and help with training efficiency. We achieve this by embedding their learning within an option-based framework without compromising performance.

## 1 Introduction

Multi-agent reinforcement learning (MARL) has recently received considerable recognition due to its broad applicability to many challenging real-world problems. Such applications include autonomous vehicles (Zhou et al., 2021), industry automation (Hanga & Kovalchuk, 2019), robot swarm control (Hüttenrauch et al., 2019), and robot manipulation (Orr & Dutta, 2023). However, due to the highly challenging nature of multi-agent problems, its scalability to these real-world applications has been limited. From one perspective, each agent's probability transition function is correlated through their independently updated policies, making the environment fundamentally non-stationary from the perspective of each agent and only gets exacerbated when more agents are involved. From another perspective, if we consider agents as joint learners, the exponential growth of their action-observation space in proportion to the number of agents would make algorithmic applications to multi-agent problems infeasible and unscalable.

Recent MARL work, including value-based (Sunehag et al., 2018; Rashid et al., 2018; Son et al., 2019; Wang et al., 2021a) and policy-based (Yu et al., 2022) methods, have concentrated on a technique to alleviate the mentioned issues coined *parameter sharing and decentralized execution* (Oliehoek & Amato, 2016), where all agents share a decentralized network, representing either a value or policy, learned using a shared parameter set. The following strategy has garnered interest due to its simplicity, reduced complexity due to its decreased number of trainable parameters, and rapid optimization capabilities from shared experience (i.e., alleviating the non-stationarity problem). However, although approaching the multi-agent paradigm through such a

strategy has shown extraordinary successes, it principally tackles two significant components of the multi-agent paradigm, *planning* and *learning.* Through planning, agents try to abstractly model the dynamics of their environment by acting within it and use learning to refine their policy that underlies their planning capabilities. A major missing component is the *reasoning* capabilities of agents. Incorporating proper reasoning approaches into the MARL framework would enable approaches for specialized action execution over specific time scales, leading foreseeably to a possibility for the reuse of knowledge.

The concept of knowledge reuse is a general approach within the MARL paradigm to achieve model generalization throughout tasks (Da Silva & Costa, 2019), that is, through transfer from one task to another. Such generalizations may either be network-based or embedding-based methods. Network-based methods concentrate on designing universal architectures capable of extracting both source and target task knowledge (Wang et al., 2020b). Embedding-based methods employ similarity measures of tasks to capture specific dynamics (Qin et al., 2023; Schäfer et al., 2023). In contrast to generalizing directly throughout tasks, an alternative knowledge reuse approach concentrates on task-independent sub-task structures (Wang et al., 2020a; 2021b; Yang et al., 2022). A tangent approach to task decomposition is temporal abstraction, specifically through the use of *options* (Sutton et al., 1999). Within MARL, options can be seen as certain courses of action that facilitate the learning process and knowledge reuse when compared to classical task decomposition methods. Specifically, the options framework allows agents to operate over extended time scales, enables them to discover useful sub-policies and behaviors, and promotes specialization and coordination. Motivated by these principles, we hypothesize that approaching MARL following the options framework can lead not only to more robust and efficient agents with optimized decision-making processes, but also to more diverse behaviors that would not be found simply using primitive actions. These are advantages that cannot be easily achieved when using direct task decomposition.

In this sense, this work presents a unique framework that extrapolates temporal abstraction via options by learning dynamic intra-option mixtures (DIOMIX) within cooperative MARL. Our work proposes to introduce the options framework within the Dec-POMDP paradigm, approaching the multi-agent problem via options without pre-defining them from prior domain knowledge. The per-agent architecture encodes two representations: (1) an option representation, also called the policy-over-options, that is underlined by an option-value function learned from the trajectory of each agent, and (2) an option-action value function that acts as an intra-option policy conditioned specifically on the option provided by the policy-over-options. To increase the representational capacity of our option representations, we optimize them against an advantage learning scheme by replacing the usually temporal difference objective found with action-value functions to advantage functions instead. Moreover, to reduce the risk of intra-option policies being led towards one-step options (i.e., directly executing primitive actions), we propose a regularization approach on the option-value functions, increasing the feasibility of extended specialized temporal action execution.

We assess our approach by employing the environmental testbed offered by JaxMARL (Rutherford et al., 2023), utilizing both SMAX, a newly developed micromanagement task set designed to emulate StarCraft II, and the multi-particle environment (MPE). We perform our empirical analysis around SMAX and use MPE to further demonstrate the effectiveness of our approach. We additionally perform ablation studies to see whether the regularization method used to avoid the optimization towards one-step options alleviates the concerns mentioned and efficiently helps the method in learning extended temporal actions without hindering the asymptotic performance of DIOMIX. The contribution of this work lies not in demonstrating an increase in overall performance compared to state-of-the-art temporal-abstraction/task-decomposition methods, but in the capability of learning reasonable and efficiently specialized agent-independent option sets that contain constrained action executions to specific objectives that are traceable.

## 2 Background and Preliminaries

In this work, we consider the *decentralized partially observable Markov decision process* (Dec-POMDP) (Oliehoek & Amato, 2016) with *options* (Sutton et al., 1999) to describe a *fully cooperative multi-agent task.*

**Dec-POMDPs with Options** are described as tuples: $\mathcal{M} = (n, \boldsymbol{S}, \boldsymbol{U}, \boldsymbol{O}, \mathbf{Pr}, r, \gamma, \boldsymbol{\Omega})$. $i \in \mathcal{N} \doteq \{1, \ldots, n\}$ is the set of agents within the environment. Each agent $i$ has a global state representation $s \in \boldsymbol{S}$ that describes the perfect information of the environment. Each agent can also execute actions $u^i \in U$, where

the joint action of all agents is denoted as $\boldsymbol{u} = \{u^i \mid i \in \mathcal{N}\} \in \boldsymbol{U}$. The independent partial observation received by each agent $i$ is defined as $o^i : \boldsymbol{S} \to O^i$ and is determined by the observation probability function $O(s, i) : \mathcal{S} \times \mathcal{N} \to O^i \in \boldsymbol{O}$ where the set of joint observations is denoted as $\boldsymbol{O}$. $\mathbf{Pr}$ is the state transition function which defines the probability $\mathbf{Pr}(\boldsymbol{s}' \mid \boldsymbol{s}, \boldsymbol{u}) : \boldsymbol{S} \times \boldsymbol{U} \times \boldsymbol{S} \to [0, 1]$ and $r(\boldsymbol{s}, \boldsymbol{u}) : \boldsymbol{S} \times \boldsymbol{U} \to \mathbb{R}$ is the reward function that maps the global state and joint actions to a single scalar reward. $\gamma \in [0, 1)$ denotes the discount factor that weighs the value of future rewards throughout the finite decision-making horizon. Each agent contains *Markovian* options each defined as a triplet $\omega_i = (I, \pi, \beta) \in \Omega$ where $I^{\omega_i} \subseteq \boldsymbol{S}$ denotes an initiation set, $\pi^{\omega_i} : \boldsymbol{S} \times U \to [0, 1]$ an option policy and $\beta^{\omega_i} : \boldsymbol{S}^+ \to [0, 1]$ a termination condition (see (Sutton et al., 1999) for more details). $\Omega$ denotes the set of options shared between each agent. A joint-option $\boldsymbol{\omega} = (\omega_1, \ldots, \omega_N) \in \boldsymbol{\Omega}$ denotes a vector of option components for each agent.

**Centralized Traning with Decentralized Execution (CTDE)** We employ the *centralized training with decentralized execution* (CTDE) paradigm (Oliehoek et al., 2008), where during training, each policy exploits extra global information that is available and may also share information between agents. However, during execution, each agent must act with only access to its action-observation history $\tau^i$. This paradigm is extensively used in deep multi-agent reinforcement learning. CTDE methods aim to factorize from individual action-value functions $\{Q^i(\tau^i, \cdot)\}_{i=1}^n$ a global action-value function $Q^{\text{tot}}(\boldsymbol{\tau}, \cdot)$. To guarantee the consistency between each value-functions, it is employed with an important principle known as the *individual global-max* (IGM) (Son et al., 2019) condition, represented as follows:

$$\arg\max_{\boldsymbol{u}} Q^{\mathbf{tot}}(\boldsymbol{\tau}, \boldsymbol{u}) = \left( \arg\max_{u^1} Q^1(\tau^1, u^1) \; \cdots \; \arg\max_{u^n} Q^n(\tau^n, u^n) \right) \tag{1}$$

The IGM condition informally states that maximizing the individual returns using a centralized action-value function should be equal to maximizing the returns through individual action-value functions. The implications of IGM concerning value decomposition are: (1) each agent can follow its individual function in a decentralized manner or the joint function in a centralized manner, (2) the joint function can be computed directly from the individually selected actions with respect to the individual returns of each agent.

**Factorization Structures** Throughout this work we mainly consider two factorization structures, *additivity* and *monotonicity*, proposed by VDN (Sunehag et al., 2018) and QMIX (Rashid et al., 2018), respectively. Both methods follow the general factorization structure as shown below:

$$Q^{\text{tot}}(\boldsymbol{\tau}, \boldsymbol{u}) = \Psi(Q^1(\tau^1, u^1), \ldots, Q^n(\tau^n, u^n); \mathbf{s})$$

where $\Psi = \sum_{i=1}^N Q^i(\tau^i, u^i)$ for VDN, and for QMIX the mixer $\Psi$ constrains to $\frac{\partial Q^{\text{tot}}(\boldsymbol{\tau}, \boldsymbol{u})}{\partial Q^i(\tau^i, u^i)} > 0, \forall i \in \mathcal{N}$ and is conditioned on the global state $\mathbf{s}$. It has been shown that the implementations are sufficient but not strictly necessary to satisfy the IGM constraint presented in Eq. 1. Other value-factorization methods exist, such as QTRAN (Son et al., 2019), which transforms the constraint into a linear constraint to act as a soft regularization on the overall IGM condition. WQMIX (Rashid et al., 2020) introduces a weighted mechanism into the projection of the value factorization to place more importance on higher utility joint actions. Currently, VDN and QMIX are the only provided mixing networks within our chosen experimental testbed. Therefore, we concentrate on these two value factorization methods within our work.

## 3    Related Works

**Value Factorization in MARL** The concept of value factorization within the framework of value-based MARL has been the leading cause of significant advancements in asymptotic performances over recent years. VDN (Sunehag et al., 2018) was the pioneering breakthrough that caused such progress by introducing an additivity factorization on individual $Q$-values to learn from a global $Q$-value. It was built upon IQL (Tan, 1993), which learns independent $Q$-values for each agent by treating each other agent as part of the environment. However, independently learning action-value functions changes the policies of each agent concurrently, leading to an issue known as non-stationarity, which may cause algorithms not to converge. Factorizing a global $Q$-value into individual $Q$-values was shown to alleviate this issue and avoid learning directly on the

joint action-observation space, which proves unscalable. It also introduced the learning paradigm, centralized training with decentralized execution (CTDE). More recently, CTDE approaches have arisen, such as QMIX (Rashid et al., 2018), which factorizes using a monotonicity constraint, and QTRAN (Son et al., 2019), which transforms the global $Q$-value into specific representations to increase the decomposition capacity. Other approaches, such as QPLEX (Wang et al., 2021a), propose a duplex dueling architecture to complete the IGM (Rashid et al., 2018; Son et al., 2019) function family class. Each factorization method concentrates on implementing architectures known as *mixing networks*.

**Temporal Abstraction in MARL** Although the general concept of temporal abstraction has not seen much attention within the MARL paradigm, the abstract use of the option framework Sutton et al. (1999) has previously been introduced in various works. DOC (Chakravorty et al., 2020) alleviates the combinatorial nature of Dec-POMDP and converts it into an equivalent POMDP using the notion of common information where agents share a pool of broadcasted beliefs that condition their option selection. Other works that have proposed solving Dec-POMDP with options are MacDec-POMDPs (Xiao et al., 2020), which approaches the problem by learning both a centralized and decentralized macro-action-value function and DOT (Han et al., 2019) which learn dynamic termination options by comparing each agents option selection against every other agent within the environment. Orthogonal to these learning methods, MAPTF (Yang et al., 2021) introduced a centralized approach to reuse and terminate policies for each agent by modeling multi-agent policy transfer as an option learning problem. Our work focuses on avoiding the conversion of Dec-POMDP to POMDP by directly including the option framework within it. It approaches option learning in a similar fashion to DOT; however, we avoid option communication and let each agent optimize their individual option sets.

**Task Decomposition in MARL** An important aspect of complex real-world multi-agent systems is the capability for agents to decompose tasks into sub-tasks. Enabling task decomposition allows agents to restrict themselves to specific specialized sub-tasks that are easier to learn and execute, decreasing the required overall optimization complexity of the high-level task. The easiest mechanism by which task decomposition can be leveraged is through prior knowledge whereby sub-tasks are predefined (Fosong et al., 2024). However, prior knowledge is expensive and may not be readily available in all circumstances. ROMA (Wang et al., 2020a) proposed to include roles for each agent to alleviate the use of knowledge by including a role conditioning factor on each agent's policy. Using the centralized mixer to condition itself as a role learner was also proposed in (Nguyen et al., 2022). However, this approach did not interact directly with the decentralized policy and concentrated solely on the mixing network output for centralized learning. RODE (Wang et al., 2021b) introduced a joint action space decomposition to learn roles in a pre-training phrase, conditioning each role on a subset of the joint-action set. LDSA (Yang et al., 2022) introduces dynamic subtask allocation where agents are constrained on specific abilities to handle specific subtasks through the use of a shared subtask policy interlocutor. Both roles and sub-tasks may be seen interchangeably without a loss of generality. Task decomposition may be seen as a superset of temporal abstraction with less fine-grained intricacies. In this work, we present a relaxed temporal abstraction method that can act as a task-decomposition method without loss of generality towards its advantanges.

## 4 Learning Dynamic Intra-Option Mixtures

This section introduces the DIOMIX framework as depicted in Figure 1. We first explain how we employ the Dec-POMDP with Options within our framework and introduce the assumptions we utilize. We subsequently discuss how each agent independently learns their specialized option sets in a centralized manner concerning the assumptions laid forward. Following, we demonstrate how to achieve independent representations for each option and avoid one-step options. Finally, we present the general training and inference strategy.

### 4.1 Intra-Options in Multi-Agent Reinforcement Learning

We consider multi-agent *Markov* options employed in the *call-and-return* execution model. Each agent $i$ can individually initiate an option $\omega_i$ in a state $\boldsymbol{s}$ if no option is currently executing and it is part of its initiation set $\boldsymbol{s} \in I^{\omega_i}$. An agent selects an option $\omega_i$ according to its *policy-over-options* $\pi_\Omega^i$ and then follows its *intra-option policy* $\pi^{\omega_i}$ until its termination condition $\beta^{\omega_i}$ is met. When an agent's option $\omega_i$ is executing,

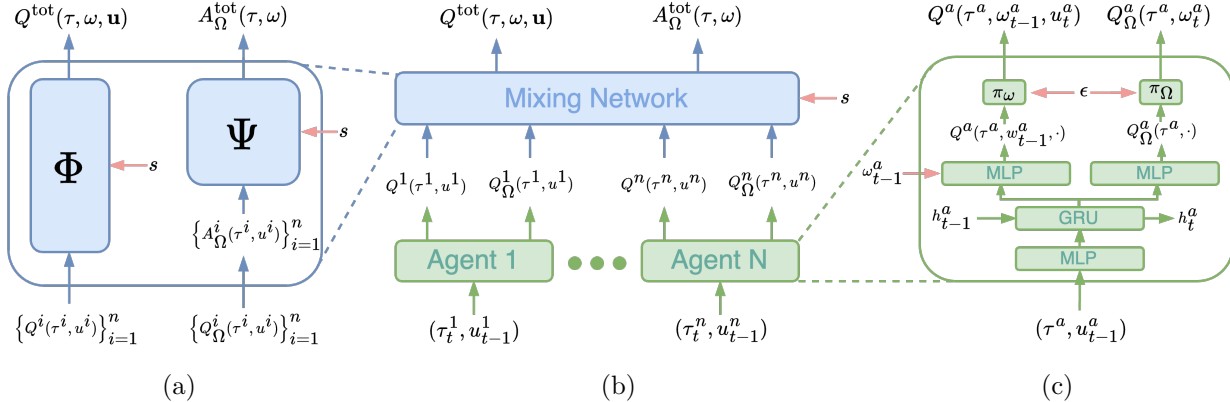

(a)     (b)     (c)

Figure 1: (a) The mixing network structure where $\Phi$ and $\Psi$ are identical network structures with independent parameter sets. The individual advantages $\{A^i_\Omega(\tau^i, u^i)\}^n_{i=1}$ are computed following the individual advantage formulation found in Definition 1 before being fed to $\Psi$. The state $\mathbf{s}$ is fed to both networks independently. (b) The overall DIOMIX architecture. (c) Agent network structure where the selected option at the previous time-step using an $\epsilon$-greedy intra-option policy $\pi_\Omega$ is fed to condition the option-action value function. The GRU output is fed to both MLP layers independently without altercations. Best viewed in color.

it generates an action $u^i$ from its option-policy $u^i \sim \pi^{\omega_i}(\cdot \mid \mathbf{s})$. Each agent may either select a one-step option, directly resulting in a primitive action $u_i$, or select an extended option that can arbitrarily execute for many time steps before terminating. We aim to avoid learning options that result in primitive actions. For ease of exposition, we assume that for each agent $i$, its options are available and executable in any given state $\forall \omega^i \in \Omega^i \; I^{w_i} = \mathcal{S}$.

Since each agent's *policy-over-options* $\pi^i_\Omega$ uses a $\epsilon$-greedy policy over each agent's option-value function $Q^i_\Omega(\mathbf{s}, \cdot)$ we eliminate the need of using a termination condition $\beta^{\omega_i}$ for each option and directly let the option-value function learn which option to select in any given state $\mathbf{s}$ indirectly acting as a termination condition. This alleviates the need to learn an independent termination network and learns to correlate the optimization of $\pi^i_\Omega$ with each option underlying *intra-option policy* $\pi^{\omega_i}$ found within each agent's option set $\Omega^i$.

### 4.2 Discovering Intra-Options using Advantage-Learning

When introducing options within the MARL learning framework, a natural question is whether each option component (i.e., policy-over-options, intra-option policies, and value functions) should all optimize on the same equal plane. Each option policy should incorporate a set of responsibilities that differ from one another, while intra-option policies should specialize on the primitive action execution that underlies the overarching specialization. Options are represented in a manner that makes them interchangeable with the learning paradigm of primitive actions. Therefore, we employ our approach in a manner whereby each component optimizes on the same plane, that is, against global rewards. However, each component optimizes its utility under its own terms. For the policy-over-options $\pi^i_\Omega : \mathcal{T}^i \to \Omega^i$, that is, the policy that receives an agent's trajectory $\tau^i \in \mathcal{T}^i$ and outputs the option $\omega^i \in \Omega^i$ to execute at the next time-step optimizes against the advantage of said option. Optimizing against the advantage function instead of directly against the option-value function inherently asks the question *how much better or worse is the executing option in comparison to all other available options on average relative to the current intra-policy?*

The change in learning objective from the classical temporal difference learning objective using the action-value or value function to the advantage-function for options motivates the reformulation of the IGM condition into an advantage-based option specialized IGM condition. The following reformulation builds upon the advantage-based IGM formalized in (Wang et al., 2021b); however, it differs in its underlying use and,

therefore, its learned representation; it also differentiates the underlying use of advantage whereby (Wang et al., 2021b) use it as a dueling concept for $Q^{\text{tot}}$ whereas we use it as a representation of $A^{\text{tot}}$ from $\{A^i\}_{i=1}^n$.

**Definition 1** (Advantage-based Option IGM). For a joint-option value function $Q_{\Omega}^{\text{tot}} : \mathcal{T} \times \Omega \to \mathbb{R}$, if there exist individual option value functions $\{Q_{\Omega}^i : \mathcal{T}^i \times \Omega^i\}_{i=1}^n$, where $\forall \boldsymbol{\tau} \in \mathcal{T}, \forall \boldsymbol{\omega} \in \boldsymbol{\Omega}, \forall i \in \mathcal{N}$ with both the joint and individual advantage of an option defined as

$$\textbf{(Joint Advantage)} \quad A^{\text{tot}}(\boldsymbol{\tau}, \boldsymbol{\omega}) = Q_{\boldsymbol{\Omega}}^{\text{tot}}(\boldsymbol{\tau}, \boldsymbol{\omega}) - V_{\boldsymbol{\Omega}}^{\text{tot}}(\boldsymbol{\tau}) \text{ where } V_{\boldsymbol{\Omega}}^{\text{tot}}(\boldsymbol{\tau}) = \max_{\boldsymbol{\omega}} Q_{\boldsymbol{\Omega}}^{\text{tot}}(\boldsymbol{\tau}, \cdot)$$

$$\textbf{(Individual Advantage)} \quad A^i(\tau^i, \omega^i) = Q_{\Omega}^i(\tau^i, \omega^i) - V_{\Omega}^i(\tau^i) \text{ where } V_{\Omega}^i(\tau^i) = \max_{\omega_i} Q_{\Omega}^i(\tau^i, \cdot)$$

such that the following holds

$$\arg\max_{\boldsymbol{\omega}} A^{\text{tot}}(\tau, \boldsymbol{\omega}) = \left( \arg\max_{\omega^1} A^1(\tau^1, \omega^1) \;\cdots\; \arg\max_{\omega^n} A^n(\tau^n, \omega^n) \right) \tag{2}$$

then, we say that the $\{Q_{\Omega}^i\}$ satisfies the advantage-based option IGM for $Q_{\Omega}^{\text{tot}}$.

From Definition 1, using the advantage-based option IGM simply constrains the regular IGM from action-value functions as described in Eq. 1 to that of the advantage-option function. However, this does not restrict the use of regular IGM on option-value functions, where one would directly optimize the individual option-value functions $\{Q^i(\tau^i, \omega^i)\}_{i=1}^n$.

We may then use the advantage-based option IGM directly on the policy-over-options by learning the option-value function through the use of a global advantage function, enabling the use of centralized training. Therefore, we apply intra-option learning through the angle of learning the advantage of any given option at any given timestep in comparison to all other available options and apply it within the classical temporal difference learning scheme as follows:

$$\mathcal{L}_{\Omega}(\Psi) = \mathbb{E}_{\mathcal{B}}\left[\left(\boldsymbol{r} + \gamma A_{\Omega}^{\text{tot}}(\boldsymbol{\tau}', \boldsymbol{w}'; \bar{\Psi}) - A_{\Omega}^{\text{tot}}(\boldsymbol{\tau}, \boldsymbol{w}; \Psi)\right)^2\right] \tag{3}$$

where $\mathcal{B}$ denotes the replay buffer. The above learning scheme is used to both learn the overall utility of any given option over a sequence of states and inherently learn whether options should continue executing or terminate indirectly learning an internal termination function.

### 4.3 Independent Intra-Option Representations

A caveat of optimizing for return is that options will inevitably shrink over time. Optimizing for return forces the option to be executed as fast as possible, possibly leading to a one-step option (i.e., a primitive action). Introducing a regularizer that prevents this shrinking can preserve the advantage of extended temporal options, which helps with the speed of learning, planning, and, importantly, reasoning. Within the option-critic (Bacon et al., 2017) framework, this approach was presented by introducing a notion of bounded rationality by using a deliberation cost (Harb et al., 2018; Bacon & Precup, 2015). The deliberation cost was used to essentially guide the agents to avoid switching options. The cost presented avoids the agent to terminate options unless another viably better option is available. In a similar vein, we require an option selection strategy that is robust in constraining the extended temporal abstraction. To achieve this constraint, we softmax the option-value function to create a distribution over the option utilities for every agent. We are then provided with a distribution over each agents available options conditioned on a trajectory as follows:

$$p_j(w_i \mid \tau) = \frac{\exp(Q_j(\tau, w_i)/\kappa)}{\sum_{w_i' \in \Omega} \exp(Q_j(\tau, w_i')/\kappa)}, \; \forall i \in |\Omega|, \forall j \in \mathcal{N} \tag{4}$$

At every timestep, each agent will independently query their policy-over-options to either terminate the currently executing option or dynamically switch to a new option. Having agents switch options at every

timestep or in an intermittent fashion would ultimately lead to learning one-step options that result in primitive actions. We, therefore, introduce a regularization scheme that aims to minimize the KL divergence of two adjacent timesteps to enhance the representations of option selection as follows:

$$\mathcal{L}_\lambda(\Psi) = \mathbb{E}_{(\boldsymbol{\tau},\boldsymbol{w})\sim\mathcal{B}}\left[\sum_j \mathcal{D}_{\mathrm{KL}}\big(\boldsymbol{p}_j(\boldsymbol{w}_j|\boldsymbol{\tau}_j) \parallel \boldsymbol{p}'_j(\boldsymbol{w}'_j|\boldsymbol{\tau}'_j)\big)\right] \tag{5}$$

where $\boldsymbol{p}'_j(\boldsymbol{w}'_j|\boldsymbol{\tau}'_j)$ is the option distribution conditioned on the trajectory starting at the next timestep onwards and $\mathcal{D}_{\mathrm{KL}}(\cdot\|\cdot)$ is the Kullback-Leibler divergence operator which calculates the relative entropy between option distribution over different timesteps. The sum is taken over each agent's option distribution differences.

## 4.4 General Training and Inference Objective

As with all value function factorization algorithms that employ the CTDE paradigm (Sunehag et al., 2018; Rashid et al., 2018; Wang et al., 2021a), our method incorporates a mixing network that maps each agent's action-value function $(Q_i, Q_{-i})$ into a global action-value function $Q_{\mathrm{tot}}$ where $Q_i$ represents the individual action-value of agent $i$ and $Q_{-i}$ the individual action-value functions of all other agents excluding agent $i$. Throughout our experiments, we explore various state-of-the-art mixing networks, namely VDN (Sunehag et al., 2018) and QMIX (Rashid et al., 2018), to evaluate which method leads to superior performance with respect to the objective we aim to achieve. However, our method is not limited to these and may extend to more complex mixing networks. We employ the temporal difference loss on the option-action value functions as follows:

$$\mathcal{L}_{\mathrm{TD}}(\theta) = \mathbb{E}_{\mathcal{B}}\big[\big(\boldsymbol{r} + \gamma \max_{\boldsymbol{u}'} Q^{\mathbf{tot}}(\boldsymbol{\tau}',\boldsymbol{\omega}',\boldsymbol{u}';\bar{\theta}) - Q^{\mathbf{tot}}(\boldsymbol{\tau},\boldsymbol{\omega},\boldsymbol{u};\theta)\big)^2\big] \tag{6}$$

where $\bar{\theta}$ are the target network parameters that are periodically updated every $n$-update steps from the objective network parameters $\theta$.

$$\mathcal{L}(\theta,\Psi) = \mathcal{L}_{\mathrm{TD}}(\theta) + \beta_\Omega \mathcal{L}_\Omega(\Psi) + \beta_\lambda \mathcal{L}_\lambda(\Psi) \tag{7}$$

where $\beta_\Omega$ and $\beta_\lambda$ are positive coefficients to weight the importance of the policy-over-option and the regularization, respectively. During the decentralized execution phase (i.e. inference) of the policy-over-option and the intra-option policy, each agent $i$ independently selects an option for the next timestep $t+1$ according to the maximum utility value of any given option for the timestep $t$, i.e. $\arg\max_\Omega Q^i_\Omega(\tau^i,\cdot)$, the selected option

---

**Algorithm 1:** Dynamic Intra-Option Mixtures

**1 repeat**
**2**      For each agent $i$, select option $\omega^i$ randomly from policy-over-options $\pi^i_\Omega$
**3**      **repeat**
**4**          Collect joint-history $\boldsymbol{\tau}_t$ from environment.
**5**          For each agent $i$, take action $u_i \sim \pi^i_\omega(\cdot \mid \tau^i)$ according to $\epsilon$-greedy option-policy $\pi^i_\omega$
**6**          Execute joint actions $\boldsymbol{u}_t$, observe reward $r(\boldsymbol{o}_t,\boldsymbol{u}_t)$ and next history $\boldsymbol{\tau}_{t+1}$
**7**          Store $(\boldsymbol{\tau}_t,\boldsymbol{\omega}_t,\boldsymbol{u}_t,r(\boldsymbol{\tau}_t,\boldsymbol{u}_t),\boldsymbol{\tau}_{t+1})$ into replay buffer $\mathcal{B}$
**8**          For each agent $i$, select option $\omega^i_{t+1}$ according to $\epsilon$-greedy policy-over-options $\pi^i_\Omega$
**9**      **until** $\boldsymbol{\tau}_{t+1}$ is terminal;
**10**      Sample a random minibatch of $\mathcal{K}$ samples from $\mathcal{B} : \{(\boldsymbol{\tau}_k,\boldsymbol{\omega}_k,\boldsymbol{u}_k,r(\boldsymbol{\tau}_k,\boldsymbol{u}_k),\boldsymbol{\tau}_{t+1})\}_\mathcal{K}$
**11**      Update $\{\theta_i\}_{i=1}^N, \Psi$ according to (7)
**12**      Update target networks $\bar{\boldsymbol{\theta}} \leftarrow \boldsymbol{\theta}, \bar{\Psi} \leftarrow \Psi$
**13 until** convergence;

is used to condition the intra-option policy which selects a primitive action under the selected option, i.e., $\arg\max_u Q^i(\tau^i, \omega^i, \cdot)$. Algorithm 1 provides the pseudocode that demonstrates our approach.

# 5 Experiments

In this section, through a series of experiments, we explore the properties and behavior of DIOMIX by investigating the following questions: (1) Can DIOMIX learn *independent* dynamic intra-options and assign said options adequately? (Sec. 5.3.1) (2) Can the learned intra-options specialize by restricting the overall action space into separate sub-action spaces (Sec 5.3.1) (3) Does the proposed regularization method attribute to extended temporal action execution? If so, are the learned options more specialized, and does the overall performance of our method get affected? (Sec. 5.3.2) (4) How does the asymptotic performance compare against the chosen baseline and does the learn options/sub-tasks representations differ between our approach and the baseline? (Sec. 5.2).

## 5.1 Experimental Setup

**Environment** We evaluate DIOMIX on both the SMAX and MPE benchmarks. In our experiments, we adopt the default environment settings provided by JaxMARL for SMAX. We consider various scenarios supplied by the SMAX benchmark that are classified into three different difficulty categories: *Easy*, *Hard*, and *Super Hard* scenarios.

**Baselines** We use LDSA (Yang et al., 2022), VDN (Sunehag et al., 2018) and QMIX (Rashid et al., 2018) as baselines. Other tasks-based decomposition methods such as ROMA (Wang et al., 2020a) and RODE (Wang et al., 2021b) exist and provide impressive results in task decomposition for MARL both implicitly and explicitly. However, LDSA has been shown to provide the *best* empirical results and significantly improves the overall asymptotic performance compared to other task-based decomposition algorithms. We follow the JaxMARL approach of comparing the environmental median test returns instead of the median win rates. Further details are in Appendix A.

## 5.2 Asymptotic performances

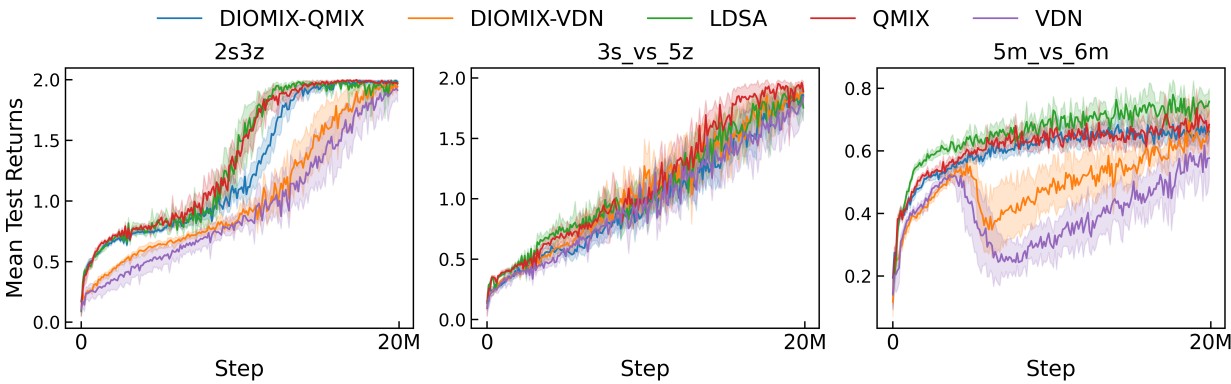

Figure 2: Comparison of our method DIOMIX against baselines on both *Hard* and *Super Hard* SMAX scenarios: `2s3z`, `3s_vs_5z`, and `5m_vs_6m`. The solid line shows the median test return across 10 seeds, and the shaded regions correspond to the 25% and 75% percentiles.

We compare the asymptotic performance of our method DIOMIX against the state-of-the-art task-decomposition method LDSA and the two barebones mixers used within our method on the SMAX benchmark (Rutherford et al., 2023) found in Figure 2 and on MPE demonstrated on Figure 3. Figure 2 shows the learning curves of our method against these baselines. DIOMIX performs practically identically to LDSA in all tested environments `2s3z`, `3s_vs_5z`, and `5m_vs_6m`. Given the similarity in performances, we further analyzed the learned subtask distribution of LDSA and its selection process for each examined task and see

that although the performances are quite similar, LDSA optimizes only one sub-task distribution every time. This demonstrates that compared to LDSA, our method is able to surpass its representational capabilities by inherently enabling the trace of the decision outcomes taken by an agent and comparing why an option was selected compared to another (as demonstrated in Sec. 5.3.1). We also compare our method using both VDN and QMIX as a mixer network and clearly see the intended outcome on SMAX, where QMIX surpasses the representational capacity of both the option-action-value and option-value functions.

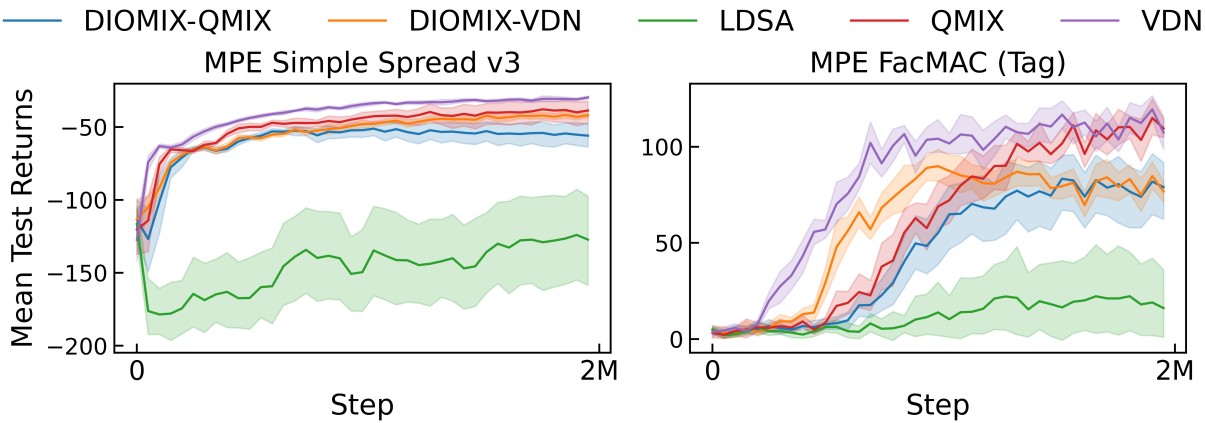

Figure 3: Comparison of our method DIOMIX against baselines on two MPE tasks: `Spread` and `Tag`. The solid line shows the median test return across 10 seeds, and the shaded regions correspond to the 25% and 75% percentiles.

We perform additional experiments on MPE using the same baselines as above and compare the overall performance achieved. As with SMAX, we follow the indicated metrics to compare against those provided by JaxMARL and, therefore, compare against the median test return on two MPE tasks: `Spread` and `Tag`. The version of `Tag` used is an extension of *Simple Tag* that was proposed in (Peng et al., 2021) as a fully cooperative version of predator-prey. Given that our method is presented for the discrete action space cases, we convert the default continuous action space into a discrete one within the JaxMARL codebase. Figure 3 demonstrates the results achieved. Our method interestingly surpasses the performances of LDSA, being the closest baseline to our method on both tasks. LDSA seems incapable of learning optimal policies, whereas DIOMIX is able to achieve close results in comparison to its no-option learning counterparts, VDN and QMIX. These experiments demonstrate the generalizability of our method in contrast to LDSA.

## 5.3 Analysis

### 5.3.1 Dynamic Intra-Option Assignments

This section analyzes the dynamic intra-option assignment performed independently by each agent's policy-over-options during one episode as shown in Figure 4. For this investigation and to enable a detailed analysis, we perform this exploration on the SMAX scenario `3m`. The `3m` scenario is the simplest task contained within the SMAX benchmark and consists of 3 ally and 3 enemy marines. Since our objective here is to demonstrate how DIOMIX separates and specializes its available actions into specialized options executed over various time scales, it provides a preliminary visualization of the assignment properties of our method.

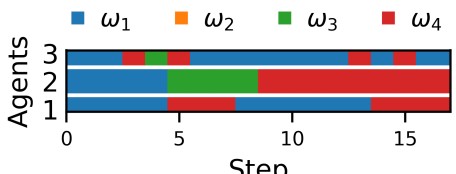

Figure 4: Dynamically learned intra-options for each agent in one test episode on `3m`. Each color indicates a different learned option selected by $w_j \sim Q_\Omega^i(\tau^i, \cdot) \ \forall i \in \mathcal{N}, \forall j \in \Omega$.

A few notable insights can be extracted from this simplistic scenario. We can see that throughout the entire episode, out of 4 possible learned options, DIOMIX has only concentrated on specializing $\omega_1, \omega_3$ and

$\omega_4$, inherently avoiding the extra computational burden of specializing an entire whole other time scaled action execution sequence. Additionally, we can see that for agent 3, the regularization method did not completely avoid one-step options, where $\omega_4$ was constrained to directly executing a primitive action and then terminating. Each time agent 3 executed the option $\omega_4$, it executed a different moving action (i.e., 1 - 4), which supports the insight that these options have shrunk to one-step options.

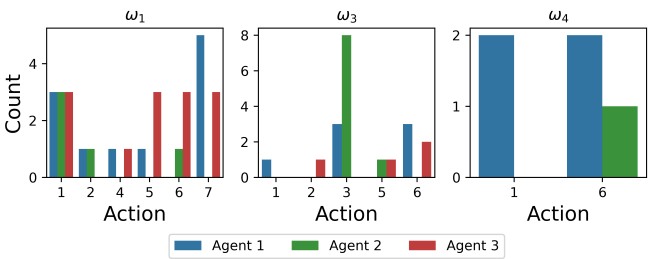

Figure 5: Independent action execution count per option grouped by agent for one episode. Represents the dynamic intra-option execution of Figure 4.

To understand the intricacies of these options, how they specialize, get executed, and differentiate from one another at every time step, we analyze the proportion of actions executed by each option for each agent. Figure 5 showcases the frequency of each executed primitive action within each option for each agent. Since each agent tries to learn options independently, we assume that agents learn completely different option representations and specialize their options differently. Figure 5 confirms our assumption.

For example, agent 1 only specializes 2 out of 4 possible options, namely $\omega_1$ and $\omega_4$. Option $\omega_4$ specializes in attack and retreat actions by executing only the attack and retreat actions. As for agent 2, we can see that option $\omega_3$ principally specializes as a movement-only option, which could suggest an option that either fully retreats after attacking or being attacked or the opposite.

### 5.3.2 Regularization Effects on Option Specialization

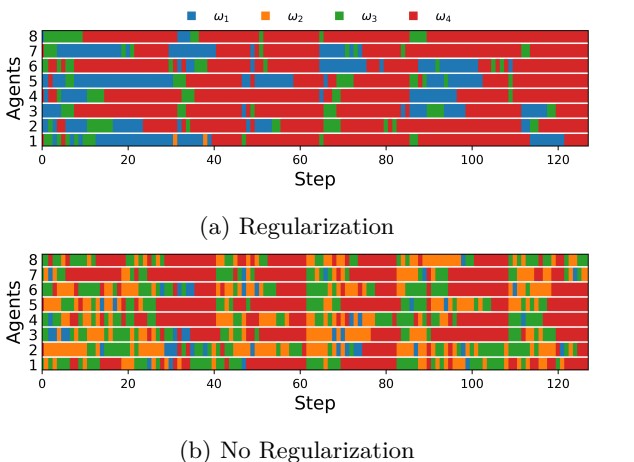

(a) Regularization

(b) No Regularization

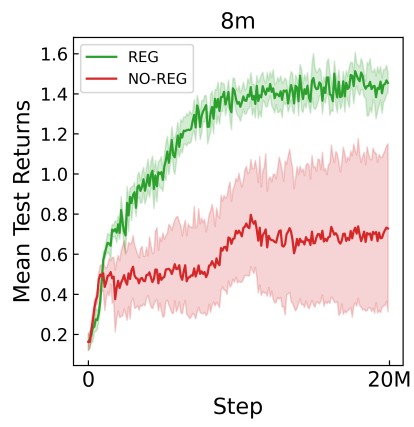

(c) Asymptotic performances comparison

Figure 6: The effects of the proposed regularization method on option specialization. Figures (a)-(b) show the comparison between the learned options using regularization (a) and without regularization (b). Figure (c) shows the test mean return between using the regularization to complement (a) and without regularization to complement (b).

Removing the regularization method proposed introduces two significant downfalls to our proposed approach, DIOMIX, as shown in Figure 6. The first is the irregularity and inefficiency of the learned options. Figure 6b clearly demonstrates this irregularity where each agent optimizes to execute more one-step options where options essentially become primitive actions. We can also see that without $\mathcal{L}_\lambda(\Psi)$, not only do options shrink, but the general training objective tries to overcompensate the learning onto one option (e.g. in this specific example $\omega_4$) making each other option useless as extended temporal actions. In Figure 6b, $\omega_4$ seems to have been the only option to have learned a specialization objective. The second downfall is the performances achieved in Figure 6c without regularization, which complements the findings found in

the option execution plot Figure 6b. Without option regularization, DIOMIX has an extremely hard time adjusting to the objective it has to learn.

### 5.3.3 Advantage-based vs Action-Value Learning on Options

An interesting question is whether the advantage-based centralized learning scheme actually enhances the learning capability of option-value functions, in contrast to using the classical IGM condition. We provide this ablation by comparing learning with and without advantage-based centralization (i.e. with only the option-value function). As we can see from Figure 7, using advantage-based learning with options provides much higher asymptotic performance with significantly less variance compared to using the classic IGM condition. This is because using the actual value function without advantage does not pose the question as to how much better executing an option is compared to all other options available and learning with respect to that instead of directly learning which option leads to the highest utility at that given timestep.

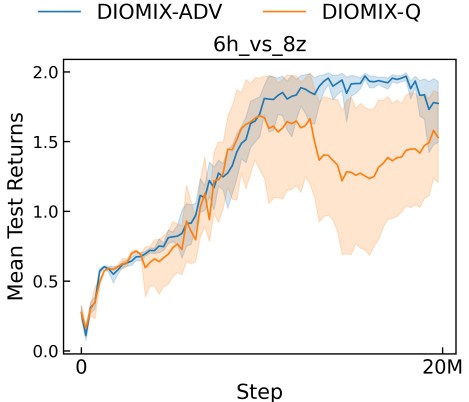

Figure 7: Comparing optimizing *policy-over-options* using $A^{\mathbf{tot}}(\boldsymbol{\tau}, \boldsymbol{w})$ versus $Q^{\mathbf{tot}}(\boldsymbol{\tau}, \boldsymbol{w})$.

## 6 Conclusion

What constitutes a *good* set of options, sub-goals, or sub-tasks is a question that remains very much open and an active area of research within single-agent reinforcement learning. However, the multi-agent reinforcement learning field has not seen much active research within this area. Therefore, we propose using temporal abstraction via options and demonstrate how agents can deal with specializing extended temporal action execution over various time scales. The empirical results suggest that through a centralized learning approach, our method learns effective and efficient temporal abstractions independently for each agent. The learned representations are traceable, specialized, and robust. Through this work, we provide an orthogonal perspective on task decomposition through temporal abstraction, contributing a valuable approach to the field's ongoing development. Advancing this field is crucial as it can lead to more sophisticated, adaptable, and efficient multi-agent systems, which are increasingly relevant in complex, real-world applications. While the concept is fundamentally challenging and not yet fully solved, our research highlights its potential and underscores the importance of continued investigation in this promising area.

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

# A Experimental Details

Our experiments were performed on a desktop machine with an 18-core Intel Core i9-10980XE CPU, 256 RAM, and 3 NVIDIA A6000 GPUs. Each experiment was run with 8 parallel environments and a single GPU for training. The experiments were executed using JaxMARL 0.0.2 under Jax 0.4.25 and Flax 0.8.1.

Using JaxMARL as our environment testbed limits the number of available algorithms due to the shift from PyTorch (Paszke et al., 2019) to Jax (Bradbury et al., 2018). Comparing PyMARL (Samvelyan et al., 2019) implementations under SMAC (Samvelyan et al., 2019) against JaxMARL implementations under SMAX would lead to an inaccurate comparison. Therefore, we run VDN and QMIX using the provided baseline implementations found directly within the JaxMARL codebase and also provide a one-to-one re-implementation of LDSA under Jax within our code release for an impartial baseline.

## A.1 Hyperparameters

<table>
<tr><td colspan="2">Table 1: Hyperparameters for SMAX</td><td colspan="2">Table 2: Hyperparameters for MPE</td></tr>
<tr><th>Hyperparameter</th><th>Value</th><th>Hyperparameter</th><th>Value</th></tr>
<tr><td>num envs</td><td>8</td><td>num envs</td><td>8</td></tr>
<tr><td>num steps</td><td>128</td><td>num steps</td><td>25</td></tr>
<tr><td>buffer size</td><td>3000</td><td>buffer size</td><td>5000</td></tr>
<tr><td>buffer batch size</td><td>32</td><td>buffer batch size</td><td>32</td></tr>
<tr><td>total timesteps</td><td>$2 \times 10^7$</td><td>total timesteps</td><td>$2.05 \times 10^6$</td></tr>
<tr><td>agent hidden dim</td><td>256</td><td>agent hidden dim</td><td>64</td></tr>
<tr><td>agent option dim$^\dagger$</td><td>4</td><td>agent option dim$^\dagger$</td><td>4</td></tr>
<tr><td>agent init scale</td><td>1.0</td><td>agent init scale</td><td>2.0</td></tr>
<tr><td>epsilon start</td><td>1.0</td><td>epsilon start</td><td>1.0</td></tr>
<tr><td>epsilon finish</td><td>0.05</td><td>epsilon finish</td><td>0.05</td></tr>
<tr><td>epsilon anneal time</td><td>100000</td><td>epsilon anneal time</td><td>10000</td></tr>
<tr><td>mixer embedding dim*</td><td>64</td><td>mixer embedding dim*</td><td>64</td></tr>
<tr><td>mixer hidden dim*</td><td>256</td><td>mixer hidden dim*</td><td>256</td></tr>
<tr><td>mixer init scale*</td><td>0.001</td><td>mixer init scale*</td><td>0.001</td></tr>
<tr><td>max grad norm</td><td>10</td><td>max grad norm</td><td>10</td></tr>
<tr><td>target update interval</td><td>200</td><td>target update interval</td><td>200</td></tr>
<tr><td>lr</td><td>0.001</td><td>lr</td><td>0.001</td></tr>
<tr><td>eps adam</td><td>0.00001</td><td>eps adam</td><td>0.00001</td></tr>
<tr><td>weight decay adam</td><td>$1 \times 10^{-6}$</td><td>weight decay adam</td><td>$1 \times 10^{-6}$</td></tr>
<tr><td>gamma</td><td>0.99</td><td>gamma</td><td>0.99</td></tr>
<tr><td>num test episodes</td><td>32</td><td>num test episodes</td><td>32</td></tr>
<tr><td>test interval</td><td>$1 \times 10^5$</td><td>test interval</td><td>$1 \times 10^5$</td></tr>
</table>

($\dagger$ parameters specific to DIOMIX, * parameters specific to QMIX)

