# OpenReview forum: "DIOMIX: A Dynamic Multi-Agent Reinforcement Learning Mixing Structure for Independent Intra-Option Learning"
_TMLR — Withdrawn by Authors_

### Review · Reviewer_oiax · 2024-07-10

**Summary Of Contributions:**

This work proposes Dynamic Intra-Options Mixtures (DIOMIX) for cooperative Multi-Agent Reinforcement Learning (MARL), integrating options as temporal abstractions into value function factorization algorithms within the Centralized Training with Decentralized Execution (CTDE) paradigm.

In general, the paper follows the common practice of value factorization approaches in cooperative MARL. Regarding options, the authors assume any option initiates everywhere and eliminates the need of termination function by making high-level decision for each time-step. The authors propose using advantage instead of option-value function. To prevent option collapse / shrinking, the authors propose a regularization method to prevent options shrinking into one-step actions (this contrasts with previous work, which used deliberation costs to address the issue).

The experiment compare the proposed algorithm against several common baselines in cooperative MARL, e.g., VDN, QMIX.

**Audience:**

Yes

**Broader Impact Concerns:**

Enhanced reasoning and decision-making capabilities could be leveraged in ways that may adversely affect individuals or groups, especially in critical domains like healthcare, finance, or autonomous systems.

**Claims And Evidence:**

No

**Requested Changes:**

Please see the weaknesses above.

**Strengths And Weaknesses:**

## Strengths
The section Background and Preliminaries is well-written and easy to follow, and the related work is comprehensively summarized.

The idea of incorporating temporal abstraction in MARL setting is reasonable.


## Weaknesses
### Methodology
In Definition 1, the $V$ value is defined as the maximum of the $Q$ value rather than the expectation as usually appeared in literature. Also, $\arg \max Q = \arg \max A$ result in the same action regardless of $V$. Since the action selection directly depends on $Q$ as described in the following sections, it is unnecessary to define this IGM.

In Eq.3, this temporal difference objective semantically learns an on-policy Q-value function rather than the advantage, despite the symbol $A^\text{tot}$. In addition, off-policy data sampled from the replay $\mathcal{B}$ should be used to train an on-policy loss.

Regarding the number of options, is it fixed or learned? Does "option dim" mean "option number" in Table 1?

How does Eq.5 prevent options from shrinking? For option execution, you used eps-greedy which for most of the time takes the max-valued option. There can be a counterexample that two Boltzmann distributions are close but with different max values, or two Boltzmann distributions are far away but with the same max values. More importantly, optimizing Eq.5 may in general lead to another kind of collapse: all actions in one specific option (which may also minimize the loss).



### Experiment
The motivation of the paper is a bit unclear. Why do you want to use options, and what is the benefit? The authors claim that "such temporal abstraction helps agent specialization, task simplification, and help with training efficiency, leading to explainable reasoning capabilities", but no strong evidence is shown to achieve these desiderata.

Firstly, in Figures 2 and 3, the DIOMIX variants of QMIX and VDN are not clearly better than the vanilla methods, which in contrast with "help with training efficiency".

For 5.3.1, no curve showing the convergence. How do we know whether the performance converges when you test the agent? For Fig.4, one test episode may not be enough to understand the overall behavior.

What is the performance of learning option-value function rather than learning the advantage? As this is one of your contributions, it's better to report this result.

In Sec. 5.2, the authors evaluated on three SMAC tasks, namely one easy tasks (2s3z), and two hard tasks (3s_vs_5z, 5m_vs_6m) [1,2]. However, recent work has shown that open-loop policies, which are conditioned only on the timestep, can achieve  performance indistinguishable from closed-loop policies across many SMAC scenarios including hard tasks such as 3s5z. This suggests that these SMAC tasks are not demonstrating challenges of Dec-POMDP. To strongly support their claims, I suggest the authors to evaluate on more hard/super-hard tasks. I also recommend SMACv2 [3] which consists of tasks with sufficient stochasticity.

In Sec. 5.2, the authors also evaluated on two MPE tasks (Spread and Tag). However, baseline methods VDN and QMIX can easily achieve strong performance in these tasks [4]. Fig.3 also shows indistinguishable learning curves. I recommend hard/super-hard SMAC tasks and SMACv2 tasks.

### Writing
In intro, "an option representation, also called the policy-over-options" may be misleading. In convention, representation seems more like a feature vector while policy-over-options map the current state to an option.

There are many terms that do neither follow the convention nor are well-defined. For example, the authors mention "Intra-Options" several times, which may not be canonical in the option literature. Do you mean intra-option learning, or intra-option policy?

In section 4.3, the authors mention, "A caveat of optimizing for return is that options will inevitably shrink over time," which may not always hold true. As there are (infinite) many set of options that can achieve optimal policy, the learned options may not be the ones desired but do not necessarily shrink to primitive actions. The statement "Optimizing for return forces the option to be executed as fast as possible, possibly leading to a one-step option" is also imprecise, as maximizing return does not inherently prompt the option to be executed more quickly (e.g., always sticking to a single specific option can also achieve optimality).

For Eq.4, why not stick to the notation in Definition 1 ($Q^j_\Omega$)?

---

[1] Wen, Muning, et al. "Multi-agent reinforcement learning is a sequence modeling problem." Advances in Neural Information Processing Systems 35 (2022): 16509-16521.

[2] Hu, Jian, et al. "Rethinking the implementation tricks and monotonicity constraint in cooperative multi-agent reinforcement learning." arXiv preprint arXiv:2102.03479 (2021).

[3] Ellis, Benjamin, et al. "Smacv2: An improved benchmark for cooperative multi-agent reinforcement learning." Advances in Neural Information Processing Systems 36 (2024).

[4] Papoudakis, Georgios, et al. "Benchmarking multi-agent deep reinforcement learning algorithms in cooperative tasks." arXiv preprint arXiv:2006.07869 (2020).

---

### Review · Reviewer_BDiz · 2024-07-11

**Summary Of Contributions:**

This paper presents DIOMIX: an options-based MARL algorithm in the vein of CTDE Q-learning methods like QMIX.

There are two distinctive features of DIOMIX:
- Use of an advantage-based target (rather than value-based) for the option selection policy
- Use of a regularisation term to prevent the shrinking of the option horizon

The paper presents experiments showing that DIOMIX does not perform much worse than the QMIX and VDN baselines. It then presents analysis of three aspects of the algorithm, aiming to answer:
1. whether different options correspond to distinct behaviours
2. whether introduced temporal regularisation approach is successful at reducing the shrinking of the option horizon
3. whether the advantage-based target for the policy-over-options provides benefits over a value-based target

**Audience:**

No

**Claims And Evidence:**

No

**Requested Changes:**

# Critical Changes
- Clarify options definition
- Run analysis experiments on all relevant tasks rather than just one. Report additional results in supplementary materials if necessary.
- Run and report hyperparameter tuning procedure to ensure result differences are not just due to hyperparameters.
- Fix action labelling in SMAX (Sec 5.3.1): actions "1,2,3,4,5,6,7" are not interpretable and should be replaced by their meanings (e.g. "up", "down", "shoot", etc)
- Build histograms for Fig 5 across many episodes, not just one.
- Provide more insight into the explainability of the learned options. E.g. can you explain all options for all agents? Can you show that forcing agents to choose specific options causes them to perform a specific role?
- Reframe "shrinking options" problem or provide evidence that options are shrinking over time without regularisation (as opposed to getting longer over time with regularisation)
- Make sure all claims in abstract/introduction are in line with results

# Improvement Changes
- (Strongly recommended): Overcooked experiments
- Experiments forcing certain options and observing the resulting behaviour would be more convincing evidence of specialisation than the current discussion in 5.3.1.
- Fig 5 is difficult to interpret. Seems easier to interpret if each subplot corresponded to one agent, and each colour corresponded to one option.
- Provide more justification for computing KL divergence between softmax policies.
- Consistent use of "intra-option policy" (rather than option-policy, which appears twice)
- In my experience, the JaxMARL suite is very efficient, and it seems feasible that the authors could increase from using 10 seeds per experiment to using 128 seeds without incurring significant
- Consider using similar colours to visually group lines on plots (e.g. DIOMIX-QMIX in red and QMIX in orange, since these are united by their basis on QMIX)

**Strengths And Weaknesses:**

# Strengths

The use of options in multi-agent reinforcement learning is an interesting direction which deserves more investigation.

The chosen baselines seem reasonable.

The paper is well-structured.

# Weaknesses

General comment: in my opinion this paper currently does not meet the bar for correctness. I'd also like to see some deeper insights into the functioning of the approach in order before publication in this venue — though the general topic may be of interest to TMLR's audience, at present I think the findings themselves may not be.

## Background/Related Work/Method

The definition of Dec-POMDPs with Options becomes difficult to follow when options are introduced: perhaps chiefly because the index $i$, which is already used to index agents in the Dec-POMDP, is seemingly reused to label options. Thus it appears like there is one option per agent. The notation in this section should be clarified.

The paper claims that "CTDE methods aim to factorize from individual action-value functions ... a global action-value function". It seems that this applies to only a subset of CTDE methods rather than all CTDE methods.

The paper claims that roles and sub-tasks are interchangeable wlog, although this is not entirely clear to me: it seems like the definition of roles from Wang et al 2020a and Wang et al 2021b, and subtasks from Yang et al 2022 and Fosong et al 2024 are not exactly equivalent. In particular, notions of subtasks seem to typically allow for differences in transition dynamics, whereas roles do not.

In section 4.2 it's not clear what "all optimize on the same equal plane" means — this should be made more precise.

Definition 1: It seems like the definition of advantage used here bakes in an assumption that a greedy policy is used. This is not a problem at present, because an $\epsilon$-greedy policy-over-options is used. But this is possibly worth commenting on in the paper.

In section 4.3, DIOMIX minimises the KL divergence between two successive softmax policies. While this seems reasonable, it perhaps requires more justification because it seems that the DIOMIX does not use a softmax policy to actually chose the option, but rather an $\epsilon$-greedy one. There's no reason in principle why the KL divergence between the successive $\epsilon$-greedy policies couldn't be chosen.

## Experiments

While other works have used SMAC to test role assignment, it's not clear that this is an environment well-suited to role/option decomposition. The authors should strongly consider running experiments in Overcooked (also available in JaxMARL) which has a clearer need for temporally-extended options than SMAX or MPE.

It's unclear why more environments haven't been chosen for each experiment. Focussing on SMAX, the asymptotic performance results use 2s3z, 3s_vs_5z, 5m_vs_6m. The option assignment experiment (5.3.1) uses the 3m task only; the regularisation experiments (5.3.2) uses the 8m task only; and the advantage-vs-value ablation experiment (5.3.3) uses the 6h_vs_8z task only. It's not clear why this is the case, and readers could assume it is due to cherrypicking. It can be OK to focus on one environment in a given experiment if results are consistent across other environments — but at present, it is not clear whether e.g. the DIOMIX-Q performs worse than DIOMIX-ADV across all tasks or just 6h_vs_8z.

Winrate results in SMAX can be informative, and should be included in supplementary material.

Though hyperparameters are reported in the appendix, there is no discussion of hyperparameter tuning. In order to be confident that results are reflecting real differences between algorithms rather than simply being artifacts of hyperparameter choice, hyperparameter sweeps should be performed and the procedure reported. For example, it seems plausible that choosing different hyperparameters for DIOMIX-Q could improve performance.

Option assignment: Figure 5 is sampled from one short episode. There are very few samples involved, so it seems difficult to draw solid conclusions based on the reported counts. It seems best if these histograms were built over many episodes instead (of course the networks used to generate each episode should be the same).

Is the temporal issue with options really one of "shrinking"? It seems to me like it's more likely to be a problem of "not growing" and that by default with 4 options you'd expect a 75% chance of switching to a new option in the next timestep with a randomly initialised network. If the problem is indeed one of "shrinking" then there are no experiments which establish this fact (e.g. experiments showing that over time options get shorter without regularisation).

It's not clear why 4 options were chosen. How does performance change when 2 or 8 options are chosen? What about the extreme case of 1 option — is performance the same as QMIX/VDN?

I don't find the discussion about the reason advantaged-based learning performs better convincing. Is the argument different to the standard variance-reduction argument for advantage-based metrics?

## Unsupported Claims
**Reasoning**: At various points in the paper, DIOMIX is motivated by enabling the addition of reasoning capabilities to MARL-trained agents. However, the intuition behind this is unclear, as options seemingly have little to do with reasoning. Furthermore, the experiments do not clearly demonstrate that DIOMIX is capable of reasoning, nor that DIOMIX is more capable of reasoning than the baseline methods.

**Traceability**: the paper does not convincingly show that the learned options pertain to "specific objectives that are traceable". The main evidence presented for this claim can be found in sec 5.3.1. However, this evidence is weak (see my earlier comments) and in any case only suggests two specific objectives. It is difficult for the readers to verify that for agent 1 $\omega_4$ really does correspond to an attack/retreat stance given that we observe only 4 actions taken under $\omega_4$; and it is not stated what agent 1's $\omega_1$ corresponds to in terms of specific objective.

**Agent Specialisation**: the abstract claims DIOMIX can lead to agent specialisation, which I take to mean that agents , but no evidence is provided for this.

**Training Efficiency**: the abstract claims DIOMIX helps with training efficiency, but the only relevant evidence would address a narrower claim of "DIOMIX helps improve training efficiency relative to LDSA but not QMIX/VDN".

**Diversity**: Section one presents the hypothesis that "following the options framework can lead not only to more robust and efficient agents with optimized decision-making processes, but also to more diverse behaviors that would not be found simply using primitive actions". This hypothesis is not tested in this work, and not intuition is given, though it is in fairness presented as a hypothesis. This is followed by "These are advantages that cannot be easily achieved when using direct task decomposition" which is a claim that is also not established here.

---

### Review · Reviewer_fZT1 · 2024-07-15

**Summary Of Contributions:**

The authors present a framework, DIOMIX, for incorporating option learning and mixing into a multi-agent reinforcement learning agent. DIOMIX (Dynamic Intra-Options Mixtures) attempts to learn a set of distinct options for each agent, which they refer to as the agent's intra-option policies, that are then selected from using each agent's individual policy-over-options.

The authors assume that options are usable from any given state (i.e., their initiation set is the entire state space). They also advocate for the agent's policy-over-options to select the currently running option based on a softmax over the concurrently learned option value functions (equation 4). Notably, this means that options can be initiated and terminated at any given tilmestep.

An agent's options are learned based on an advantage-based prediction loss (equation 3). To encourage agents to learn multi-step options, they also introduce a KL regularization loss that encourages the policy-over-options to be close between time steps.

Empirical evaluation is done using environments from the relatively new JaxMARL library, specifically a discretization of the SMAX environment (a simplified version of SMAC, the StarCraft Multi-Agent Challenge) and the multi-particle environment (MPE). Performance is compared with algorithms that factorize the team's value function into per-agent value functions methods, specifically value-decomposition networks (VDN), QMIX, and LDSA.

**Audience:**

Yes

**Broader Impact Concerns:**

I have no concerns about the ethical implications of the work that would demand a Broader Impact Statement.

**Claims And Evidence:**

No

**Requested Changes:**

## Critical changes

The authors must address the main weaknesses mentioned to secure my recommendation:
- pointing out the commonalities between DIOMIX and LDSA and providing proper attribution,
- noting the connection to bandits literature,
- explaining how the MPE environment was discretized, and
- providing a more detailed and statistically robust analysis of the options generated by DIOMIX in section 5.3. For example, in Figure 5, specify what the different action indices mean. Are the differences consequential, or vacuous? Many domains have "different" actions that you can take that largely don't impact outcome.

## Changes to strengthen the work

**Section 1**: The authors discuss learning, planning, and reasoning, noting the importance of reasoning and that it is a major missing component. The meaning of learning and planning are expanded upon, but not reasoning. It would help crystallize the authors' intention by being more precise with what they mean here.

**Dec-POMDP notation**: There are some places where per-agent and joint versions (e.g., observations, options) are difficult to distinguish. For example,
> $O^{i} \in \mathbf{O}$ where the set of joint observations is denoted as $\mathbf{O}$.

This would suggest $O^{i}$ is a joint observation, but is only a single agent's observation. Similarly there is $\Omega$ and $\mathbf{\Omega}$, which are very hard to visually distinguish.

**Other math notation**: In Equation 3, for example, there is some prime (e.g., $\tau'$, $\omega'$) and bar notation ($\bar{\Psi}$) that I don’t see introduced/explained. The intention can be inferred after further reading, but it is not obvious at first glance.

**Equation 5**: Is $p’$ or $\omega’$ really needed? Isn’t it the same p just conditioned on the different trajectory? Also, why would this be for the next timestep onwards? Isn't this just the KL between adjacent time steps?

**Section 5.1** More details about the SMAX and MPE benchmarks would help readers contextualize results. While readers can see referenced work for some details, the present details are not really sufficient to understand the experiments without prior knowledge. Furthermore, the authors note:
> We follow the JaxMARL approach of comparing the environmental median test returns instead of the median win rates.

However, I did not see that reflected in the JaxMARL paper. Both it and the LDSA paper seem to present median win rates. I would advise consistency with those results, or an explanation/motivation for the difference. Figures 2 and 3 also show "Mean Test Returns" for the y-axis, not the median. This should be corrected so the text agrees with figures.

**Section 5.1**:
> Further details are in Appendix A.

Make it clear what is in Appendix A. It is not clear from the text that this is where your network and algorithm parameters are specified.


### Minor typos/edits (not critical, just to strengthen work)

**Section 1, paragraph 4**:
> In this sense

In what sense?

**Section 1, paragraph 4**:
> replacing the usually temporal

replacing the usual temporal

**Page 4**:
> to complete the IGM.

Complete? Do you mean "satisfy"?

**Strengths And Weaknesses:**

## Strengths

The authors note several challenges in scaling MARL techniques to sophisticated domains and present some useful background works that have received attention in the area.

## Weaknesses

My biggest concern about the paper is that there are several elements that seem very similar to parts of the LDSA paper. At a surface glance, there is strikingly similar phrasing between the first paragraphs of each paper's introduction. Examining the opening paragraph of each paper's experiment section, there is a set of questions that is different in content but incredibly similar in structure. More importantly, many of DIOMIX's algorithmic choices are echos of those in LDSA. For example, LDSA also uses a KL divergence regularization term between subtask selection distributions on subsequent time steps. This is, at best, lacking acknowledgement and attribution by DIOMIX's authors.

Additional related work is also not mentioned. DIOMIX's approach is very reminiscent of online learning literature such as multi-armed bandits, particularly given the authors' assumption that options can be initiated at any state and no termination states need to be learned. With DIOMIX using each option effectively as a full policy (i.e., an "expert"), plus the advantage-based learning framing, this work appears to have clear unmentioned parallels to the regret notions common to bandit literature (e.g., the Exp4 algorithm). Readers would benefit from being aware of the connection to this area of research.

The authors central contribution appears to be grounded in the following statement:
> The contribution of this work lies not in demonstrating an increase in overall performance compared to state-of-the-art temporal-abstraction/task-decomposition methods, but in the capability of learning reasonable and efficiently specialized agent-independent option sets that contain constrained action executions to specific objectives that are traceable.

While the authors provide some analysis to show DIOMIX is competitive with some other baselines, experiments examining the nature of the options that were discovered felt thin. A lack of experimental details and rigorous analysis left me unconvinced and worried about reproducibility. For example, section 5.3 examined the options, but only appeared to provide examples of options used on a single episode. This seems insufficient to demonstrate that this behaviour is consistent across different episodes or training runs. Moreover, there is no real comparison of option quality between DIOMIX and other methods. The LDSA paper, for example, seems to similarly demonstrate different subtasks getting selected in Figure 6. Notable details about experiments also were omitted, such as how the MPE environments were discretized, or what the action indexes in Figure 5 mean in terms of agent behaviour.

Finally, I believe the authors could improve the flow of the paper. There are several places where it seems the authors begin referring to a concept before really explaining their meaning, only to later expand. This was particularly evident in formal notation. For example, in equation 6, $\bar{\theta}$ is used before ever mentioning that a target network would be used (also in a near-identical was to LDSA). This often left me wondering if I had missed something. It would help flow to guide readers in advance.

---

### Note · Authors · 2024-07-28

I have read and agree with the venue's withdrawal policy on behalf of myself and my co-authors.